# Facile Preparation of a Novel Vanillin-Containing DOPO Derivate as a Flame Retardant for Epoxy Resins

**DOI:** 10.3390/ma15093155

**Published:** 2022-04-27

**Authors:** Liping Chen, Zhonglin Luo, Biaobing Wang

**Affiliations:** Jiangsu Key Laboratory of Environmentally Friendly Polymeric Materials, Jiangsu Collaborative Innovation Center of Photovoltaic Science and Engineering, School of Materials Science and Engineering, Changzhou University, Changzhou 213164, China; chenliping526@163.com (L.C.); zhonglinluo@cczu.edu.cn (Z.L.)

**Keywords:** epoxy resin, bio-based, flame retardancy, lower phosphorus content, mechanism

## Abstract

A novel bio-based flame retardant designated AVD has been synthesized in a one-pot process via the reaction of 9,10-dihydro-9-oxa-10-phospha-phenanthrene-10-oxide (DOPO), vanillin (VN), and 2- aminobenzothiazole (ABT). The structure of AVD was confirmed using Fourier transform infrared spectroscopy (FTIR), and ^1^H and ^31^P nuclear magnetic resonance spectroscopy (NMR). The curing process, thermal stability, flame retardancy, and mechanical properties of the epoxy resin (EP) modified with AVD have been investigated comprehensively. The extent of curing, the glass transition temperature and the crosslinking density of the blend decreased gradually with increasing AVD content. The thermogravimetric analysis (TGA) was used to demonstrate that the presence of AVD reduced the thermal decomposition rate for EP and enhanced the formation of carbon residue during resin decomposition. A blend of 7.5 wt% AVD (0.52% phosphorus) displays a UL-94V-0 rating and a LOI of 31.1%. Reduction of the peak heat release rate, total heat release rate and total smoke production was 41.26%, 35.70%, and 24.03%, respectively, as compared to the values for pure EP. The improved flame retardancy of the flame retardant epoxy (FREP) may be attributed to the formation of a compact and continuous protective char layer into the condensed phase as well as the release of non-combustible gases and phosphorus-containing radicals from the decomposition of AVD in the gas phase. AVD is a new and efficient biobased flame retardant for epoxy with great prospects for industrial applications.

## 1. Introduction

Epoxy resin (EP), as an important thermosetting resin which displays characteristics of chemical resistance, low curing shrinkage, outstanding adhesion, and great electrical insulation. It has been widely applied in aerospace, coatings, adhesives, and microelectronics [1,2,3]. However, intrinsic flammability restricts its applications in many high-tech areas. It is accordingly urgent to improve the flame retardancy of EP [4,5]. Traditional halogen-containing flame retardants (FRs) produce toxic gases such as dioxins during combustion, which are hazardous to human health and also have a negative effect on the environment [6]. Consequently, halogen-containing FRs have been forbidden in many applications, and the development of halogen-free FRs has become imperative. Recently, 9,10-dihydro-9-oxa-10-phospha-phenanthrene-10-oxide (DOPO) has become a promising phosphorus-based FR due to its non-toxic properties and high phosphorus content. Nevertheless, a satisfactory flame retardant effect is achieved only at high DOPO content in epoxy resin [7,8,9]. The presence of DOPO at these levels caused a deterioration of the mechanical properties of the resin. Fortunately, the highly reactive P-H bond of DOPO permits the introduction of other flame-retardant elements into its molecular structure, including nitrogen [10,11], sulfur [12,13], silicon [14,15], boron [16,17], etc. These DOPO-based derivatives display better flame retardant efficiency for EP due to cooperative effect of multiple flame retardant elements.

To reduce dependence on petroleum resources and to reduce toxicity, new biobased flame retardants are being rapidly developed [18]. Some biobased materials have received much attention for the preparation of epoxy resin flame retardants, such as vanillin [19], isosorbide [20], tartaric acid [21], chitosan [22], furans [23], crop-based phenolics [24], glycerol/adipic acid hyperbranched poly (ester)s, [25] etc. All of the above biobased raw materials have great potential for application in the development of biobased phosphorus flame retardants. Daniel et al. [26] converted renewable isosorbide into the corresponding diacrylate and used it to synthesize four phosphorus-containing compounds that showed good flame retardant properties in epoxy resins. Two of them are stable at temperatures close to 400 °C and may be suitable flame retardants for polymers processed at high temperatures. Among them, vanillin is a promising starting material for the preparation of flame retardants due to the high reactivity of its aldehyde and the presence of phenolic hydroxyl group. A biobased reactive FR (VDG) has been synthesized in a one-pot reaction involving DOPO, vanillin, and 3,5-diamino-1,2,4-triazole flame retardant (VDG). A high LOI value of 37.0% and UL-94 V-0 rating were observed for the cured EP system containing 2.0 wt% VDG. This material also exhibits antibacterial effects toward *E. coli*. [27].

It has previously been demonstrated that the N/S-containing thiazole has great potential for the construction of efficient FRs [28,29,30]. A novel bio-based FR (marked as AVD) has synthesized from the reaction of DOPO, vanillin, and 2-aminobenzothiazole. The chemical structure of AVD was characterized using Fourier transform infrared spectroscopy (FTIR), and ^1^H and ^31^P nuclear magnetic resonance spectroscopy (NMR). EP/AVD thermosets with different AVD loading levels were produced and their curing behaviors, thermal stability, flammability and combustion behaviors were evaluated. Moreover, a flame retardant mode of action for AVD in EP has been proposed.

## 2. Experimental

### 2.1. Materials

Epoxy resin with the epoxy value of 0.51 mol/100 g (commercial name: E-51) was purchased from CNOOC Changzhou Coating Chemical Research Institute Co., Ltd. (Changzhou, China). DOPO, vanillin (VN, 99%), 4,4′-Diaminodiphenyl methane (DDM, 98%) were acquired from Aladdin Reagents Co., Ltd. (Shanghai, China). 2-Aminobenzothiazole was purchased from Jiangsu Qiangsheng Functional Chemical Co., Ltd (Nanjing, China). Absolute ethanol was supplied by Sinopharm Chemical Reagent Co., Ltd (Shanghai, China). All raw materials were not purified and were used directly.

### 2.2. Synthesis of AVD

The AVD was one-pot synthesized, and the synthetic route is illustrated in Figure 1. To a 250 mL three-necked flask equipped with magnetic stirrer and reflux condenser, vanillin (VN) (0.06 mol, 9.129 g), 2-aminobenzothiazole (ABT) (0.06 mol, 6.0084 g) and anhydrous ethanol (100 mL) were added. After reacting at 80 °C for 5 h, DOPO (0.06 mol, 9.129 g) was added and stirred continuously for another 12 h. The crude product was collected by filtration, washed three times with anhydrous ethanol, and then the product was dried to constant weight in a vacuum oven at 70 °C. The pale-yellow powder was obtained (Yield: 76%, melting temperature: 205 °C).

### 2.3. Preparation of EP and Flame Retardant EPs (FREPs)

The mole ratio of the amino group to epoxy group was 1:1 for all samples, and the formulations were shown in Table 1. Firstly, a transparent EP/AVD solution was obtained under magnetic stirring at 130 °C, and then cooled to 90 °C. Afterwards, DDM was introduced and kept stirring until it was completely dissolved. The mixture was then dumped into the preheated silicone rubber mold and cured at 100 °C for 2 h and 150 °C for 3 h. The Pure EP/DDM thermoset was prepared with the above-mentioned procedure.

### 2.4. Characterization

FTIR spectra were obtained using a Perkin Elmer instrument (Waltham, MA, USA) over a spectral range of 4000–400 cm^−1^. All samples were milled with KBr and pressed into tablets.

^1^H NMR and ^31^P NMR spectra were collected on a Bruker Advance III-500 NMR spectrometer (Bruker, Waltham, MA, USA) using the DMSO-d_6_ as deuterated solvent.

Differential scanning calorimetry (DSC) analysis was carried out on a Perkin-Elmer DSC 8000 (PE, Waltham, MA, USA) at different heating rate from 30 °C to 250 °C under N_2_ atmosphere. The weight of all samples was fixed at about 5 mg.

Thermogravimetric analysis (TGA) was performed on a Perkin-Elmer TGA 4000 (Waltham, MA, USA) with a nitrogen flow rate of 20 mL·min^−1^. The specimen (about 8 mg) was placed in an alumina crucible and heated from 30 to 700 °C at a heating rate of 10 °C·min^−1^.

The vertical burning (UL-94) test was measured by a CZF-3 instrument (Shine Ray Instrument Co. Ltd., Nanjing, China) according to ASTM D3801 standard.

The limited oxygen index (LOI) value was measured using an HC-2 oxygen index meter (Jiang Ning Co. Ltd., Nanjing, China) according to ASTM D2863.

The combustion behaviors were tested on a FTT cone calorimeter (Fire Testing Technology, East Grinstead, UK) according to the ISO 5660-1 standard at an external heat flux of 35 kW·m^−2^. Three replicates with dimensions of 100 × 100 × 3 mm^3^ and weight of about 36.5 g were tested, and their average values were collected as each point data.

The microscopic morphologies of residual char were observed by a SUPRA55 scanning electron microscope (SEM) with an acceleration voltage of 5 kV.

X-ray photoelectron spectroscopy (XPS) was determined by an ESCALAB 250Xi system (Thermo Fischer Scientific, Waltham, MA, USA), using Al Kα excitation radiation (hν = 1486.6 eV).

Raman spectroscopy was collected through a DXR2xi laser Raman spectrometer (LRs) (Thermo Fischer Scientific, Waltham, MA, USA) in the range of 500–3000 cm^−1^ with an excitation wavelength of 532 nm.

TG-IR spectroscopy was conducted using a combination system of a TGA 4000 thermogravimetric analyzer and a Spectrum II FTIR spectrophotometer. The sample (around 20 mg) was heated from 30 to 700 °C at 10 °C·min^−1^ with a nitrogen flow rate of 20 mL/min.

A dynamic mechanical analysis (DMA) was carried out on a Perkin-Elmer DMA8000 (PE, Waltham, MA, USA). A three-point bending mold with an amplitude of 20 μm and a frequency of 1 Hz was selected. The experimental temperature interval was 30–260 °C at a rate of 10 °C·min^−1^. (Dimensions of all samples: 40 × 6 × 3 mm^3^).

## 3. Results and Discussion

### 3.1. Characterization of AVD

The FTIR spectra of the target product (AVD) and raw materials (VN, ABT, and DOPO) are presented in Figure 1. With respect to the FTIR spectrum of AVD, the absorption peak at 3413 cm^−1^ is assigned to the stretching vibration of -OH in VN, and the disappearance of the characteristic absorption peak of -CHO observed at 1667 cm^−1^ indicates a complete reaction between VN and ABT [28]. Meanwhile, the typical P-H stretching vibration absorption peak of DOPO at 2437 cm^−1^ [29] disappears in the spectrum of AVD, and the double peaks (3396 cm^−1^ and 3272 cm^−1^) of -NH_2_ in ABT shifts to a single peak (3227 cm^−1^) of -NH in AVD [31]; the peak at 1377 cm^−1^ is ascribed to the stretching vibration of C-N [29]. All of these phenomena confirm that the addition reaction between the Schiff-base intermediate and DOPO proceeds successfully. Additionally, the absorption peaks at 1598 cm^−1^, 1449 cm^−1^, 1238 cm^−1^, and 1210 cm^−1^ are attributed to the benzene ring, C=N in the thiazole ring, P=O, and P-O-C stretching vibrations, respectively [28]. The above results confirm the initial chemical structure of AVD.

Both ^1^H NMR and ^31^P NMR spectra were performed to further check the chemical structure of AVD. As shown in the ^1^H NMR spectrum of AVD (Figure 2a), the chemical shifts at 5.66 and 5.80 ppm are attributed to the hydrogen atom on the chiral carbon attached to the DOPO group [31]. The chemical shifts at 6.66–6.75 ppm, 6.85–8.18 ppm, and 9.05 ppm are assigned to N-H, p.roton hydrogen on the benzene ring (Ar-H), and the signal of -OH, respectively. The integral area ratio of the different proton chemical environments is in agreement with the theoretical values. Furthermore, the ^31^P NMR spectrum of AVD (Figure 2b) presents two signal peaks at 28.78 and 30.25 ppm. It indicates that the P element in AVD is in two different chemical environments, which might be due to the presence of the chiral carbon atom. Based on the above analysis, it is concluded that the target product AVD was synthesized successfully.

### 3.2. Curing Behaviors

To investigate the effect of incorporation of AVD on the curing process of epoxy resin, the non-isothermal curing kinetics of the epoxy systems at different heating rates were performed by DSC, and the resultant DSC curves are shown in Figure 3a–d. As can be seen, the $T_{P}$ values of all samples shift toward higher temperatures as the heating rate increases. Moreover, the $T_{P}$ values gradually become greater with increasing AVD content at the same heating rate. This effect is mainly due to the steric hindrance of the rigid groups such as DOPO and benzothiazole in the AVD structure reduces the reactivity of ring-opening curing of epoxy resin.

The apparent activation energy (Ea) of the epoxy systems are further calculated according to the Kissinger’s (Equation (1)) and Ozawa’s methods (Equation (2)) [32] and the fitted curves of $\ln\left(\beta /T_{P}^{2}\right)$ and lnβ versus $1/T_{P}\times 10^{3}$ are illustrated in Figure 3e,f, and the results are summarized in Table 2.
(1)$$\ln\left(\beta /T_{P}^{2}\right)=\ln\left(\mathrm{AR}/E_{a}\right)-E_{a}/{\mathrm{RT}}_{P}$$
(2)$$\ln\beta =\ln\left({\mathrm{AE}}_{a}/R\right)-1.052E_{a}/{\mathrm{RT}}_{P}-5.331$$
wherein β is the heating rate, $T_{P}$ is the curing peak temperature, A is the pre-exponential factor and R is the ideal gas constant (8.314 J K^−1^ mol^−1^).

All of the Ea values calculated from both Kissinger’s and Ozawa’s methods increase with the increasing of AVD content. It suggests that the addition of AVD enhances the energy barrier of the curing reaction, which further verifies the presence of the steric hindrance.

### 3.3. Thermal Stability

The thermal stability of the pure EP and FREPs under nitrogen atmosphere was evaluated by TGA. The corresponding TG and DTG curves are depicted in Figure 4, and the related data are summarized in Table 3. Obviously, the AVD gives lower T_5%_ (272.8 °C) and greater CR_700_ (30.1%) than the pure EP. With the addition of AVD, the T_5%_ and T_max_ of the cured FREPs decrease with the increase of AVD content, which suggests that the AVD promotes the decomposition of the epoxy matrix on advance. However, the R_max_ decreases from 18.8%·min^−1^ for pure EP to 10.7%·min^−1^ for FREP-10, indicating that the presence of AVD delays the decomposition of the EP at a higher temperature. Meanwhile, the FREP-10 sample gives a CR_700_ of 24.5%, which is greater than that of the pure EP (19.7%). It implies that the incorporation of AVD improves the carbon formation ability of the cured FREPs.

### 3.4. Flame Retardancy of EP and FREPs

LOI and UL-94 measurements were carried out to assess the flame retardancy, and the resultant LOI values and UL-94 rating are listed in Table 4. Apparently, the pure EP is a combustible polymer with the LOI value of 25% and fails to pass the UL-94 vertical burn testing. With the incorporation of 5 wt% AVD, the FREP-5 sample presents an LOI value of 30% and UL-94 V-1 testing. Furthermore, the FRSP containing 7.5 wt% AVD (the P content is 0.52 wt%) achieves the LOI value up to 31.3% and UL-94 V-0 rating. The results reveal that the AVD has high flame retardant efficiency for epoxy resin.

### 3.5. Analysis of Fire Behaviors

The cone calorimetry test (CCT) is one of the most effective methods used in the laboratory to evaluate the combustion behavior of materials [33,34]. This method can provide a series of important parameters about material flammability, including time to ignition (TTI), peak heat release rate (PHRR), total heat release (THR), total smoke release (TSP), fire growth rate index (FIGRA), average effective combustion heat burn (av-EHC), average CO yield (av-COY), average CO_2_ yield (av-CO_2_Y) and char residue after combustion, all of which are summarized in Table 5. Figure 5 depicts some important curves, which include the heat release rate (HRR), total heat release (THR), total smoke release (TSP), smoke release rate (SPR), CO_2_ production rate, and residue mass over time.

As compared with the pure EP, the FREPs have greater TTI values which tend to increase with the increasing AVD content. This can be ascribed to the fact that the presence of AVD advances the earlier decomposition of the EP matrix. This result is consistent with that of the TGA test.

The heat release rate (HRR) is one of the key parameters to assess the burning intensity. As shown in Table 5, the PHRR and THR values of the pure EPare 1452.5 kW·m^−2^ and 67.3 MJ·m^−2^, respectively. The combustion intensity decreases significantly with increasing AVD content. For instance, the PHRR and THR values are 657.6 kW·m^−2^ and 57.4 MJ·m^−2^ for the FREP-10 sample, which are reduced by 54.7% and 14.7% with comparison to the pure EP, respectively. This demonstrates that the incorporation of AVD can effectively suppress the combustion intensity of epoxy composites. In addition, the fire growth rate index (FIGRA) was commonly used to assess the rate of fire growth during combustion and calculated based on HRR curves according to Equation (3) [35].
(3)$$\mathrm{FIGRA}=\mathrm{PHRR}/T_{\mathrm{PHRR}}$$

The lower FIGRA value of the material indicates the higher fire safety performance. The results in Table 5 display that the FIGRA value decreases significantly after the incorporation of AVD, from 11.2 kW·m^−2^·s^−1^ for pure EP to 3.44 kW·m^−2^·s^−1^ for FREP-10, with a reduction of 69.2%. Therefore, it can be concluded that the FREPs containing AVD have excellent fire safety.

It is well known that smoke is the cause of death for the majority of victims who lose their life due to respiratory injuries in fires. Therefore, the smoke suppression performance of flame retardant materials is a critical parameter. Obviously, the TSP value of FREP-10 (20.4 m^2^) is reduced by15.4% as compared that of the pristine EP (24.1 m^2^), illustrating that the AVD displays good smoke suppression on the epoxy resin.

Furthermore, the average effective heat of combustion (av-EHC, HRR/MLR) is an essential parameter to measure the degree of combustion of volatile substances in the gas phase. Seen from Table 5, the av-EHC value [36] of the FREPs is decreased gradually with the increasing AVD loading level, indicating that AVD has a good gas-phase flame retardant effect. It is also shown in Table 5 that the FREPs display reduced av-CO_2_Y values and increased av-COY values. This is attributed to the occurrence of incomplete combustion, further verifying the gas-phase flame retardant effect of AVD. Moreover, much more carbon residuals were left for FREPs, confirming that the incorporation of AVD promotes the carbonization of the EP. This might be attributable to the fact that the decomposition of the compounds containing DOPO can generate polyphosphates which catalyze the dehydration and esterification of the EP matrix, thus facilitating the formation of a carbonaceous protective layer.

### 3.6. Morphology of Char Residues

Figure 6 displays the digital photographs and SEM images of char residues after CCT. As shown in the images, the pure EP was burned almost completely and left a few broken and loose char residues (Figure 6a_2_). However, with the incorporation of AVD, much more and continuous char residues (Figure 6b_2_–d_2_) with greater expansion height (Figure 6b_1_–d_1_) were obtained after CCT. It is further evident from the SEM images that the pure EP exhibits a thin and friable char layer with many cracks on the surface and inside (Figure 6a_3_), which completely fails to protect the underlying substrate. Conversely, compact and continuous char layers (Figure 6b_3_–d_3_), which effectively isolate the underlying substrate from heat and oxygen, are observed for FREPs. This is mainly due to the fact that the phosphoric acid from the decomposition of AVD catalyzes the dehydration and carbonization of the EP substrate, and the benzothiazole group with better thermal stability also contributes to the production of char residues.

### 3.7. Chemical Component of Residual Char

XPS was applied to analyze the compositional changes of the char residues of pure EP and FREPs. The XPS spectra with possible peak positions are presented in Figure 7, and the corresponding elemental contents are listed in Table 6. Compared with the neat EP, the FREP-7.5 sample displays a greater ratio of C/O and N/O. It means that the char layer of FREP-7.5 is rich in nitrogen heterocycles and aromatic compounds. Moreover, a low oxygen content is also found for FREP-7.5. The decrease of oxygen content is mainly due to the formation of PO∙ and PO_2_∙, which volatilize into the gas phase to play a flame retardant role. Furthermore, the presence of P and S elements in the char residues indicates that they can act as the flame retardant mode of action in the condensed phase.

Figure 8 shows the C1s, N1s, O1s, and P2p spectra of the char residue of FREP-7.5. In Figure 8a, the C1s spectrum is decomposed to three bands at 284.3 eV (aliphatic and aromatic C-H and C-C), 286.0 eV (C-O-C and P-O-C) and 288.3 eV (carbonyl) [37]. In the N1s spectrum (Figure 8b), the bands at 397.7 eV and 399.4 eV are ascribed to C-N or P-N and N-H on the amine group, respectively [38]. For the O1s spectrum (Figure 8c), the peaks at 531.5 eV and 532.8 eV are attributed to C=O/P=O and -O- in the C-O-P group. For the P2p spectrum (Figure 8d), the P2p peak is split into two peaks at 132.4 eV and 133.4 eV, which are assigned to the P-O-C group in phosphate and P=O, respectively [39]. The above results suggest that the AVD can decompose to produce phosphate-containing compounds which promotine the dehydration and esterification of the EP substrate to form a high-quality protective layer.

### 3.8. Raman Characterization of the Char Residues

Raman spectroscopy can be applied to characterize the degree of graphitization of carbon materials. Figure 9 plots the Raman spectra of the char residue of pure EP and FREPs. The spectra of all samples have two peaks belonging to the D-band (around 1360 cm^−1^) and the G-band (around 1600 cm^−1^), which represent disordered carbon and graphitized carbon, respectively [40]. The value of A_D_/A_G_ (area ratio of D-band to G-band) can reflect the graphitization degree of the charcoal residue, and its lower value means the higher graphitization degree of the corresponding char layer. The A_D_/A_G_ values of the char residues from pure EP, FREP-5, FREP-7.5, and FREP-10 were determined to be 2.57, 2.50, 2.30, and 2.11, respectively. It can be seen that the values of FREPs are all lower than that of the pure EP, and the lowest value is achieved for FREP-10, indicating that the addition of AVD enhanced the graphitization of the char layer, which facilitates the formation of a denser and continuous char layer that acts as a barrier to inhibit the further degradation of the substrate.

### 3.9. Analysis of Gaseous Products of Pyrolysis of EP Composites

A TG-FTIR test was adopted to excavate the gas-phase volatiles generated during the pyrolysis of EP composites. Figure 10 demonstrated the characteristic spectra and 3D TG-FTIR spectra of the gas-phase pyrolysis products of the neat EP and FREP-7.5 at different temperatures. As can be seen, the pyrolysis product of FREP-7.5 appears earlier (349 °C) than that of pure EP (382 °C), suggesting that the earlier decomposition of the EP matrix is advanced by the introduction of AVD. Despite all this, the pyrolysis products of pure EP and flame retardant EP are almost the same at higher temperatures, including 3675 cm^−1^ (H_2_O), 2850 cm^−1^–3100 cm^−1^ (aliphatic C-H), 1337 cm^−1^ (C-N), 1252 cm^−1^ (C-O of bisphenol A), 1176 cm^−1^ (aliphatic C-O), 747 cm^−1^ (benzene C-H) [39]. With respective to FTIR spectra of FREP-7.5; nevertheless, some other bands occur at 1602 cm^−1^ (P-O-Ph), 1332 cm^−1 (^SO_2_), 1257 cm^−1^ (P=O), 1043 cm^−1^ (P-O-C), and 966 cm^−1^ (NH_3_). The variation of the peak intensities of these compounds reveals the flame retardant effect of AVD in the gas phase.

In order to confirm the flame retardant effect of AVD in the gas phase, Figure 11 depicts the variation of the spectra absorbance of combustible volatiles (hydrocarbons, aromatic compounds, carbonyl compounds, and aliphatic ethers) with time. Obviously, the intensities of the corresponding peaks are reduced with the introduction of AVD. Since combustible volatiles provide a large amount of fuel for combustion [41,42], the significant reduction of their intensities is further evidence of the radical scavenging effect of AVD decomposition products. Simultaneously, the captured aromatic compounds can be used as a carbon source, thus improving the char yield of the FREPs.

### 3.10. Potential Flame-Retardant Mode of Action

The results of the above tests indicated that AVD exerts a good flame retardant effect in both the gas phase and the condensed phase. Hence, the possible flame retardant mode of action of AVD is proposed as shown in Figure 12. AVD acts as a flame retardant in the gas phase by releasing non-combustible gases such as NH_3_, SO_2_, CO_2_, and PO∙, PO_2_∙. Non-combustible gases dilute the concentration of gases supporting combustion such as oxygen and carry away heat; phosphorus-containing radicals such as PO∙ and PO_2_∙ capture high-activity radicals (H∙ and OH∙) in the combustion area to break off the free radical chain reaction of combustion [43], and thus prevent further combustion of the matrix. Simultaneously, the decomposition of the AVD can produce polyphosphate, pyrophosphoric acid or metaphosphoric acid in the condensed phase, which can undergo an esterification reaction with the EP substrate. Dehydration and carbonization form dense char layers with a P-O-C structure. This continuous and dense char layers can block the transfer of heat and protect the EP matrix.

### 3.11. Mechanical Properties

DMA was employed to study the dynamic thermomechanical behavior of epoxy resins. Figure 13 illustrates the curves of storage modulus (E′) and tan δ with temperature for pure EP and FREPs, and the obtained results from DMA are listed in Table 7. The storage modulus at 50 °C of the FREPs are higher than that of the pure EP and increase with the increasing AVD contents. This is mainly due to the presence of rigid DOPO and benzothiazole groups of the AVD. At temperatures above T_g_, however, the AVD contents have converse influence on the storage modulus, which might be attributed to a lower crosslink density (υ_e_) of the FREPs. As can be seen in Figure 13b, the occurrence of a single peak indicates good compatibility between the AVD and EP matrix. Moreover, the T_g_ values of the FREPs are decreased with the increasing AVD contents. This is attributed to the predominance of the crosslink density over the rigid groups.

The crosslink density (υ_e_) of the cured EP can be calculated from the equation derived from the theory of rubber elasticity [44].
(4)$$\nu_{e}=\;E^{\prime }/3\mathrm{RT}$$
$E^{\prime }$: the storage modulus taken 40 °C above T_g_, R: the ideal gas constant (8.314 J K^−1^ mol^−1^), T: the thermodynamic temperature at T_g_ + 40 °C.

The calculated υ_e_ values for all samples are also summarized in Table 7. Since the presence of rigid DOPO and benzothiazole groups inhibits the motion of the molecular chains, the υ_e_ values are decreased with the increasing AVD contents.

## 4. Conclusions

A novel bio-based flame retardant AVD was one-pot synthesized using DOPO, vanillin, and 2-aminobenzothiazole as raw materials. The introduction of AVD hindered the curing process and reduced the T_g_ and crosslinking density values. The AVD showed an opposite effect on the storage modulus at temperatures above or under T_g_ due to the competition between the rigidity and lowered crosslinking density. The TGA results demonstrated that the early decomposition of AVD decreased the T_5%_ and T_max_ values of the cured FREPs while retarding the further decomposition of the EP matrix at higher temperature. The doping of AVD, EP exhibits great flame retardant properties. At a AVD loading level of 7.5 wt% (P content only 0.52 wt%), the LOI value of 31.3% and UL-94 V-0 rating were achieved for FREP-7.5. Moreover, the PHRR, THR and TSP values of FREP-7.5 declined by 54.7%, 14.7% and 15.4%, respectively. The comprehensive analysis of the char residues after CCT demonstrated that the phosphate-containing compounds dehydrated and esterified the EP matrix to form compact and continuous protective layers with high quality, which acted as physical barriers to effectively isolate the underlying substrate from heat and oxygen in the condensed phase. Additionally, the analysis of the pyrolysis volatiles showed that the release of non-combustible gases and phosphorus-containing radicals prevented the further combustion of the matrix in the gas phase. In conclusion, AVD, as an efficient and environmentally friendly bio-based flame retardant, is consistent with the concept of sustainable development and has great potential application in many fields. The exploration of more bio-based raw materials for flame retardant modification is one of the very promising development directions in flame retardant research.

## Data Availability

Not applicable.

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
