# Peer review of "Facile Preparation of a Novel Vanillin-Containing DOPO Derivate as a Flame Retardant for Epoxy Resins"

_materials, 2022, doi:10.3390/ma15093155_

Round 1

Reviewer 1 Report

In this submitted manuscript, Dr. Wang and co-authors developed a vanillin-based DOPO derivative and studied its physical and mechanical properties as a flame retardant for epoxy resins. After synthesized in one pot, this bio flame retardant (AVD) was first characterized by the FTIR and 1H and 31P NMR for structure confirmation. Then in the following thermal property studies, the relationships between the AVD contents and glass transition temperature, decomposition rate, and flame retardancy were evaluated. Meanwhile, the fire behaviors, the morphology of char residues, and the chemical component of residual char were also investigated. The potential flame-retardant mechanism was proposed to explain the flaming process.

In terms of the manuscript content and importance, the work would appeal to the broad readership of Materials and I would recommend accepting it for publication after minor revisions.

1. More background should be discussed, like providing the definitions of the LOI value and UL-94 V-0 rating and the reasons for doing such tests in section 3.4.

2. Some grammatical errors need to be corrected, for example:

1) the word “was” on the 12th line of the Abstract should be revised as “were”.

2) the sentence in the first paragraph of section 3.1, “the characteristic absorption peak of –CHO, which was observed at 1667 cm-1 for the indicates the complete reaction between VN and ABT”, is not a complete sentence and should be rewritten.

3) In the same paragraph, the sentence “…vibration of C-N, all of these phenomena…” should be separated into two sentences as “…vibration of C-N. All of these phenomena…”

3. The formats should be revised in a few places, like:

1) extra spaces were found between number/letter and dash line in “9,10- dihydro-9-oxa- 10-phospha-phenanthrene-10-oxide”.

2) The words “Figure 9” should be in bold on the second line of the first paragraph of section 3.8.

Author Response

Q 1 More background should be discussed, like providing the definitions of the LOI value and UL-94 V-0 rating and the reasons for doing such tests in section 3.4.

Reply: Thank you for your comment. The limiting oxygen index (LOI) is the volume fraction concentration of oxygen in a polymer in a mixture of oxygen and nitrogen when it can just support its combustion. It is an important parameter to characterize the combustion behavior of materials. Also, the UL-94 rating is the most widely used standard for the flammability performance of plastic materials. It is used to evaluate the ability of a material to extinguish after being ignited. These two tests provide a visual indication of the flame resistance of materials.

Q2. Some grammatical errors need to be corrected, for example:

  • the word “was” on the 12th line of the Abstract should be revised as “were”.

Reply: Thank you for your comment. We have made the corresponding modifications.

  • the sentence in the first paragraph of section 3.1, “the characteristic absorption peak of –CHO, which was observed at 1667 cm-1 for the indicates the complete reaction between VN and ABT”, is not a complete sentence and should be rewritten.

Reply: Thank you for your comment. We have revised it to “the disappearance of the characteristic absorption peak of -CHO observed at 1667 cm-1 indicates a complete reaction between VN and ABT.”

3) In the same paragraph, the sentence “…vibration of C-N, all of these phenomena…” should be separated into two sentences as “…vibration of C-N. All of these phenomena…”

Reply: Thank you for pointing this out. We have made changes in the text in the appropriate places.

Q3. The formats should be revised in a few places, like:

  • extra spaces were found between number/letter and dash line in “9,10- dihydro-9-oxa- 10-phospha-phenanthrene-10-oxide”.

Reply: Thank you for pointing this out. We have deleted the extra spaces.

2) The words “Figure 9” should be in bold on the second line of the first paragraph of section 3.8.

Reply: Thank you for your careful review. We have set "Figure 9" on the second line of the first paragraph of section 3.8. to bold.

Reviewer 2 Report

The research by Chen et al. aims to highlight the potential of new flame retardant (FR) additives for epoxy resins.

Specifically, the modification of an already validated FR, known as DOPO, with a natural substance (vanillin) is considered and the product obtained is coded as AVD.

Starting from a commercial epoxy system (epoxy pre-polymer and hardener), materials with different AVD content were studied in terms of thermal stability, flame behaviour, combustion, also verifying the effect of the new additive on the crosslinking process of the base resin.

The results obtained are very interesting and, among other things, the satisfactory flame retardant action of AVD demonstrated both in the gas phase and in the condensed phase prompted the authors to propose also a potential mechanism of action of the new additive.

Appreciating the clarity of presentation of the contents and the appropriateness of the methodologies used as well as the relevance of the new acquired knowledge useful for improving the safety of use of epoxy systems widely used for even advanced applications, only a few MINOR revisions of the text are suggested.

Paragraph 3.1 - line 4: there is something wrong with the sentence. You should probably delete "for the". Please check.

Paragraph 3.1 - lines 5-10: it is advisable to break this sentence. In particular, a "." after C-N and start a new sentence with "All of these ...".

Paragraph 3.2 - line 6: It is suggested to replace "It is mainly contributed ..." with "This effect is mainly due to ...".

Paragraph 3.1 - line 10: after the citation of the reference [31] you can insert a comma and delete the conjunction "and".

Paragraph 3.5 - line 11: it is advisable to replace "It is contributed ..." with "This can be ascribed ...".

Paragraph 3.10 - line 11: probably the word "to" after "carbonizzation" is superfluous and can be deleted. Please check.

Author Response

Q1. Paragraph 3.1 - line 4: there is something wrong with the sentence. You should probably delete "for the". Please check.

Reply: The comments get to point. We have revised in the text.

Q2. Paragraph 3.1 - lines 5-10: it is advisable to break this sentence. In particular, a "." after C-N and start a new sentence with "All of these ...".

Reply: Thanks for pointing this out. We added a "." after the C-N to break the sentence and start a new sentence with "All of those ...... ".

Q3. Paragraph 3.2 - line 6: It is suggested to replace "It is mainly contributed ..." with "This effect is mainly due to ...".

Reply: Thank you for your comment. We have replaced "It is mainly contributed ..." to "This effect is mainly due to ...".

Q4. Paragraph 3.1 - line 10: after the citation of the reference [31] you can insert a comma and delete the conjunction "and".

Reply: Thanks for pointing this out. We have inserted a comma and deleted the conjunction "and" in the Paragraph 3.1 - line 10: after the citation of the reference [31].

Q5. Paragraph 3.5 - line 11: it is advisable to replace "It is contributed ..." with "This can be ascribed ...".

Reply: Thank you for your advice. We have replaced "It is contributed ..." to "This can be ascribed ..." in the Paragraph 3.5 - line 11.

Q6. Paragraph 3.10 - line 11: probably the word "to" after "carbonization" is superfluous and can be deleted. Please check.

Reply: Thanks for pointing this out. We have deleted the word "to" after "carbonization".

Reviewer 3 Report

This manuscript reports the generation of a potential flame retardant from vanillin, 2-aminobenzothiazole and 9,10-dihydro-9-oxa-10-phenanthrene-10-oxide and assessment of its effectiveness as an additive in epoxy blends. This approach to the generation of biobased flame retardants has been widely utilized. Some key references are missing [see Insights Chem. Biochem,, 2020, 1(2) for leading references] and the discussion of the development of new biobased flame retardants is quite limited - no discussion of those derived from isosorbide, tartaric acid, chitosan, the furanics, crop-based phenolics or hyperbranched poly(ester)s. There is no mention of two with the greatest potential for commercialization, those derived from isosorbide bis-acrylate and the hyperbranched poly(ester) generated from two biomonomers, glycerol and adipic acid.

There is no indication of the purity of the additive prepared. It is a solid. Does it have a melting point? Does it display a single peak in HPLC analysis? It is suggested that the addition of P-H to a double bond reflects nucleophilic addition. That may be the case in this instance (polar bond) but is not always - it might be better to just refer to addition. It is correctly noted that values for activation energies obtained using variable temperature techniques contain contributions from several reactions which occur as the temperature changes and are only "apparent" activation energies. This should be emphasized.

Moieties containing sulfur and nitrogen are known to function as adjuvants to the action of organophosphorus flame retardants - the two function in parallel. The sulfur/nitrogen components, upon decomposition, release inert fragments to the gas phase to dilute the fuel load in the combustion zone. The nature of the acids formed during the thermal decomposition of organophosphorus compounds is strongly dependent on the level of oxygenation at phosphorus and the extent of decomposition - phosphate is generally not formed. The presence of oxygen at the surface of the degrading polymer is only important to the extent that thermooxidative processes are components of degradation. The function of a char layer at the surface of the degrading polymer is to inhibit heat feedback from the combustion zone which reduces the rate of pyrolysis to generate volatile fuel fragments. It is also well-known that thermal decomposition of DOPO releases the volatile PO radical (the work of Gaan should be cited). Solid evidence for the generation of other oxygenated phosphorus radicals is lacking. In this case, the presence of the PO radical in the gas phase is not demonstrated but rather inferred from previous results. This should be clarified.

The manuscript will require significant revision for accuracy, clarity and readability. Corrections are penciled-in directly on pages of the manuscript attached. These are illustrative of the kinds of changes needed throughout. In rewriting, careful attention should be paid to the use of articles, tenses and proper sentence structure. Superfluous phrases such as "in this work," "in the literature," etc. should be avoided. Author's names and et.al. should be omitted. References to "synergism" should be removed. No evidence for synergism is provided - in fact, the impact of the individual components of the flame retardant were not independently assessed. "Mechanism" should be "mode of action." Mechanism implies far greater molecular detail than is available here. Numerous errors need to be corrected throughout, e.g., oxygen is not a combustible gas (it supports combustion); "radical scavenging effect of AVD" should be "radical scavenging effect of AVD decomposition products"; "derived by filtration" should be "collected by filtration"; "potential flame retardant mechanism" should be "potential fame retardant mode of action"; etc. Scheme and Figure captions should be simple, explanatory and direct. For example, the caption for Scheme 1 should be "Synthesis of AVD."

Author Response

This manuscript reports the generation of a potential flame retardant from vanillin, 2-aminobenzothiazole and 9,10-dihydro-9-oxa-10-phenanthrene-10-oxide and assessment of its effectiveness as an additive in epoxy blends. This approach to the generation of biobased flame retardants has been widely utilized. Some key references are missing [see Insights Chem. Biochem, 2020, 1(2) for leading references] and the discussion of the development of new biobased flame retardants is quite limited - no discussion of those derived from isosorbide, tartaric acid, chitosan, the furanics, crop-based phenolics or hyperbranched poly(ester)s. There is no mention of two with the greatest potential for commercialization, those derived from isosorbide bis-acrylate and the hyperbranched poly(ester) generated from two biomonomers, glycerol and adipic acid.

Reply: Thank you for your comment. The discussion of bio-based flame retardants derived from isosorbide bis-acrylate was cited as reference 27.

There is no indication of the purity of the additive prepared. It is a solid. Does it have a melting point? Does it display a single peak in HPLC analysis? It is suggested that the addition of P-H to a double bond reflects nucleophilic addition. That may be the case in this instance (polar bond) but is not always - it might be better to just refer to addition. It is correctly noted that values for activation energies obtained using variable temperature techniques contain contributions from several reactions which occur as the temperature changes and are only "apparent" activation energies. This should be emphasized.

Reply: Thank you for your comment. The melting point of the prepared product was provided in the revised version. We have modified "nucleophilic addition" to "addition". The activation energy Ea represents the apparent activation energy, which is indicated in the corresponding position in the manuscript.

Moieties containing sulfur and nitrogen are known to function as adjuvants to the action of organophosphorus flame retardants - the two function in parallel. The sulfur/nitrogen components, upon decomposition, release inert fragments to the gas phase to dilute the fuel load in the combustion zone. The nature of the acids formed during the thermal decomposition of organophosphorus compounds is strongly dependent on the level of oxygenation at phosphorus and the extent of decomposition - phosphate is generally not formed. The presence of oxygen at the surface of the degrading polymer is only important to the extent that thermooxidative processes are components of degradation. The function of a char layer at the surface of the degrading polymer is to inhibit heat feedback from the combustion zone which reduces the rate of pyrolysis to generate volatile fuel fragments. It is also well-known that thermal decomposition of DOPO releases the volatile PO radical (the work of Gaan should be cited). Solid evidence for the generation of other oxygenated phosphorus radicals is lacking. In this case, the presence of the PO radical in the gas phase is not demonstrated but rather inferred from previous results. This should be clarified.

Reply: Thank you for your comment. We have analyzed the composition of volatiles by TG-IR and the telescopic vibrational absorption peaks of P=O and P-O-C appear at 1257 cm-1 and 1043 cm-1, which combined with the flame retardant mechanism of DOPO can prove the existence of PO radical to some extent. In addition, the work of Gaan is cited as a reference [42].

The manuscript will require significant revision for accuracy, clarity and readability. Corrections are penciled-in directly on pages of the manuscript attached. These are illustrative of the kinds of changes needed throughout. In rewriting, careful attention should be paid to the use of articles, tenses and proper sentence structure. Superfluous phrases such as "in this work," "in the literature," etc. should be avoided. Author's names and et.al. should be omitted. References to "synergism" should be removed. No evidence for synergism is provided - in fact, the impact of the individual components of the flame retardant were not independently assessed. "Mechanism" should be "mode of action." Mechanism implies far greater molecular detail than is available here. Numerous errors need to be corrected throughout, e.g., oxygen is not a combustible gas (it supports combustion); "radical scavenging effect of AVD" should be "radical scavenging effect of AVD decomposition products"; "derived by filtration" should be "collected by filtration"; "potential flame retardant mechanism" should be "potential flame retardant mode of action"; etc. Scheme and Figure captions should be simple, explanatory and direct. For example, the caption for Scheme 1 should be "Synthesis of AVD."

Reply: Thank you for your comment. We checked the manuscript and made changes to the use of inappropriate articles, tenses, and sentence structure. The phrase "in this work" appears only once in the text, and it is used reasonably. Also, the phrase "in the literature" does not appear in the text, so it was not revised. The word " synergism " has been revised. In addition, many researchers in this field usually use the word "mechanism" to describe the mode of action of flame retardants, but we have replaced "mechanism" with "mode of action" as you suggested. In addition, we have carefully revised the manuscript on a case-by-case basis according to your suggestion, which has been highlighted in red in the manuscript.

Reviewer 4 Report

The manuscript submitted by Chen et al. shows the preparation of an epoxy resin containing vanillin/ABT complexes, which resulted in materials with elevated fire retardancy properties. The results are clearly presented and the data corresponds to the discussions within the manuscript. This research is worth publishing and is of high relevance for related-researcher working on the topic of flame-retardant polymers. To elevate further the quality of the work, I suggest the authors reflect/comment on the following points before the article is ready for acceptance in Materials:

1) An important element to improve is to put the results obtained in the manuscript into context. How are the values shown in, for example, Table 4, compare to other fire retardant materials? Are the values shown of interest for commercial applications? Is there any room for improvement in future works? For instance, when mentioning "AVD have excellent fire safety" on page 8, is that compared to what specifically? 

2) Which specific elements of the recipe are bio-based? It is claimed as the entire flame retardant component and perhaps the entire product is bio-based (Reference to look at: 10.1039/C7NJ03776G). 

3) The correlation between the crosslink density and flame retardant property should be more clear, as this trend is important for future process optimization by polymer chemistry.

4) The authors should include pictures of the products from Table 1 to help the readers visualize the materials. 

5) Regarding FTIR, how many scans were performed? Please, state it in the manuscript.

6) Section 3.1. Each band assignment in FTIR should be together with a reference. Please, fix it in the revised version.

7)  Section 3.6. Please, refer to the figure in the text for a better understanding. 

8) Section 3.6. How do these char residues values compare to other works reported in literature where e.g. PP matrix are used?

9) Section 3.10. The suggested mechanism involves the release of non-combustible gases such as NH3, SO2, etc. The authors should suggest how to characterize these gas emissions to confirm the mechanism. For instance, why not perform some TGA coupled with GC/MS? 

10) Do the authors think the fire retardancy property could be related to the diffusivity rate of the AVD complex, which is somehow affected by the reticulation degree (Table 7)? 

11) The authors mentioned the environmentally friendly status of these materials. To improve the work from a broader environmental perspective, the authors may include a short comment on issues to be targeted in the future to increase the sustainability of these materials further. For instance, it can be referenced the use of a natural matrix to host the suggested flame retardant in this work, as previously reported in: 10.1039/C4TA04787G and https://doi.org/10.1002/adsu.202100063

Author Response

Q1. An important element to improve is to put the results obtained in the manuscript into context. How are the values shown in, for example, Table 4, compare to other fire retardant materials? Are the values shown of interest for commercial applications? Is there any room for improvement in future works? For instance, when mentioning "AVD have excellent fire safety" on page 8, is that compared to what specifically?

Reply: Thanks for your comment. The data in Table 4 show that AVD has good flame retardant effect on epoxy resin. The UL-94 rating and LOI values given in Table 4 are important parameters in determining whether this flame retardant material is commercially viable. Further reduction of flame retardant addition to achieve flame retardancy while maintaining mechanical properties is the direction of improvement in future work.

Q2. Which specific elements of the recipe are bio-based? It is claimed as the entire flame retardant component and perhaps the entire product is bio-based (Reference to look at: 10.1039/C7NJ03776G).

Reply: Thank you for your comment. One of our raw materials is a bio-based material, vanillin, which is derived from lignin.

Q3. The correlation between the crosslink density and flame retardant property should be more clear, as this trend is important for future process optimization by polymer chemistry.

Reply: Thank you for your comment. There is no direct correlation between crosslinking density and flame retardant properties. Crosslink density mainly responds to mechanical properties. The higher the cross-link density, the higher the tensile strength, but too high a cross-link degree will lead to a decrease in impact strength. At the same time, the higher the crosslink density, the better the heat resistance, but the flame retardant performance is not necessarily better.

Q4 The authors should include pictures of the products from Table 1 to help the readers visualize the materials.

Reply: Yes, it would be much better if provided the pictures of the products after UL-94 testing. Unfortunately, we had not taken pictures during testing at that time. As you know, we are temporarily unable to make new samples for better presentation due to COVID-19.

Q5. Regarding FTIR, how many scans were performed? Please, state it in the manuscript.

Reply: Thanks for pointing this out. Both background and sample were scanned 32 times. We have filled them in the characterization section of the manuscript.

Q6. Section 3.1. Each band assignment in FTIR should be together with a reference. Please, fix it in the revised version.

Reply: Thank you for your comment. We have fixed it in the revised version. The corresponding references are cited for each band assignment.

Q7. Section 3.6. Please, refer to the figure in the text for a better understanding.

Reply: Thanks, we have revised it according to your comment.

Q8. Section 3.6. How do these char residues values compare to other works reported in literature where e.g. PP matrix are used?

Reply: Thank you for your question. Most of the flame retardants used in PP matrix are intumescent flame retardants (IFR), the left char layer tends to have a very obvious expansion phenomenon. And the expansion effect of AVD on epoxy resin is lower than that of the IFR used for PP matrix.

Q9. Section 3.10. The suggested mechanism involves the release of non-combustible gases such as NH3, SO2, etc. The authors should suggest how to characterize these gas emissions to confirm the mechanism. For instance, why not perform some TGA coupled with GC/MS?

Reply: Thanks for pointing this out. TG-FTIR spectra of the gaseous volatiles confirm the release of SO2 (1332 cm-1) and NH3 (966 cm-1).

Q10. Do the authors think the fire retardancy property could be related to the diffusivity rate of the AVD complex, which is somehow affected by the reticulation degree (Table 7)?

Reply: Thank you for your comment. The flame retardant performance will be influenced by the degree of dispersion of AVD in the matrix, the better the dispersion, the flame retardant effect will also be improved. The change of crosslinking density in Table 7 can reflect the heat resistance of the material to a certain extent, but heat resistance is not equal to flame resistance, and the improved heat resistance is not necessarily better than flame resistance.

Q11. The authors mentioned the environmentally friendly status of these materials. To improve the work from a broader environmental perspective, the authors may include a short comment on issues to be targeted in the future to increase the sustainability of these materials further. For instance, it can be referenced the use of a natural matrix to host the suggested flame retardant in this work, as previously reported in: 10.1039/C4TA04787G and https://doi.org/10.1002/adsu.202100063

Reply: Thank you for your comment. We have included a short commentary in the conclusion section on the future directions that can be developed in the field of flame retardancy. And the reference (DOI: 10.1039/C4TA04787G) was cited as [18].

Round 2

Reviewer 3 Report

This manuscript is somewhat improved but is not yet ready for publication. The writing needs to be improved to clearly state what is intended. Suggestions are penciled-in directly on pages of the manuscript attached.

Author Response

Appreciate greatly for your careful revision on our manuscript. Attached PDF file please fine the revised version. 
